# Hemopatch to Prevent Lymphatic Leak after Robotic Prostatectomy and Pelvic Lymph Node Dissection: A Randomized Controlled Trial

**DOI:** 10.3390/cancers14184476

**Published:** 2022-09-15

**Authors:** Jeremy Yuen-Chun Teoh, Alex Qinyang Liu, Violet Wai-Fan Yuen, Franco Pui-Tak Lai, Steffi Kar-Kei Yuen, Samson Yun-Sang Chan, Julius Ho-Fai Wong, Joseph Kai-Man Li, Mandy Ho-Man Tam, Peter Ka-Fung Chiu, Samuel Chi-Hang Yee, Chi-Fai Ng

**Affiliations:** S.H. Ho Urology Center, Department of Surgery, Prince of Wales Hospital, The Chinese University of Hong Kong, Hong Kong, China

**Keywords:** Hemopatch, prostate cancer, prostatectomy, pelvic lymph node dissection, lymphatic leak

## Abstract

**Simple Summary:**

This trial investigated the use of Hemopatch, a novel hemostatic patch, during robotic-assisted prostate and lymph node surgery for prostate cancer. The researchers hypothesize that the use of Hemopatch could decrease lymph leak from the surgical bed, which is reflected by the drain output volume. The result shows that patients who underwent surgeries with Hemopatch had a lower drain output volume in total and per day, comparatively. In conclusion, Hemopatch use should be considered in prostate cancer surgery.

**Abstract:**

This study investigates whether the application of Hemopatch, a novel hemostatic patch, could prevent lymphatic leak after robotic-assisted radical prostatectomy (RARP) and bilateral pelvic lymph node dissection (BPLND). This is a prospective, single-center, phase III randomized controlled trial investigating the efficacy of Hemopatch in preventing lymphatic leak after RARP and BPLND. Participants were randomized to receive RARP and BPLND, with or without the use of Hemopatch, with an allocation ratio of 1:1. The primary outcome is the total drain output volume. The secondary outcomes include blood loss, operative time, lymph node yield, duration of drainage, drain output per day, hospital stay, transfusion and 30-day complications. A total of 32 patients were recruited in the study. The Hemopatch group had a significantly lower median total drain output than the control group (35 mL vs. 180 mL, *p* = 0.022) and a significantly lower drain output volume per day compared to the control group (35 mL/day vs. 89 mL/day, *p* = 0.038). There was no significant difference in the other secondary outcomes. In conclusion, the application of Hemopatch in RARP and BPLND could reduce the total drain output volume and the drain output volume per day. The use of Hemopatch should be considered to prevent lymphatic leakage after RARP and BPLND.

## 1. Introduction

In prostate cancer patients undergoing robot-assisted radical prostatectomy (RARP), the current European Association of Urology (EAU) prostate cancer guidelines recommend bilateral pelvic lymph node dissection (BPLND) for those with an estimated risk of occult nodal metastases exceeding 5% [1]. In a systematic review of 66 studies involving 275,269 patients, lymphadenectomy can identify node-positive patients who may benefit from adjuvant treatment [2].

BPLND in general is a well-tolerated procedure. However, when complications do occur, significant morbidity results. The benefits of BPLND must be carefully weighed against its potential complications. The most common complication of BPLND is lymphocoele formation. Small lymphatic vessels lack a muscular layer and adventitia, as opposed to blood vessels [3]. The transection of blood capillaries will lead to vasoconstriction and the eventual cessation of bleeding. This is not the case with small lymphatic vessels, and transection is likely to lead to prolonged lymphorrhoea. The incidence of lymphocoele varies from series to series, ranging from 0.8% to 33%, depending on the extent of lymphadenectomy, the surgical technique, the operative approach and the diagnostic approach [4,5]. Lymphocele formation could lead to abdominal or groin pain, abdominal swelling, lower urinary tract symptoms, bladder outlet obstruction, obstructive uropathy, infection, sepsis, lower extremity or genital oedema, deep vein thrombosis and even anastomotic disruption [6,7,8,9]. Moreover, lymphocele formation might also affect subsequent radiotherapy planning, if needed [10]. Prolonged lymphorrhoea lengthens hospital stay, places the patient at risk for nosocomial infection and has significant cost implications for the healthcare system [11].

Hemopatch is a haemostatic pad consisting of a collagen sheet derived from bovine dermis with an NHS-PEG (N-hydroxysuccinimide-functionalized pentaerythritol polyethylene glycol ether tetra-succinimidyl glutarate)-coated active surface. These two components act together to provide effective tissue adherence, sealing and haemostasis [12]. Upon tissue contact, NHS-PEG molecules on the active surface form covalent bonds with tissue proteins. Cross-linking NHS-PEG and proteins forms a hydrogel which acts as an effective tissue seal. Older-generation NHS-PEG products in the form of solutions of flowable sealants are quickly washed away by blood or other leaking body fluids, rendering them ineffective in the presence of active bleeding or fluid leakage. Hemopatch is a novel NHS-PEG delivery vehicle designed to overcome this limitation. Due to the open pore structure of the collagen, excess tissue fluids are readily absorbed, and the direct contact of NHS-PEG to the tissue surface can be achieved. The collagen pad is optimized to be soft, thin and pliable and has a high liquid absorption capacity. The pad is resorbed and replaced by the host tissue in six to eight weeks, with little tissue reaction.

We hypothesized that the application of Hemopatch to raw lymphatic tissue can prevent lymphorrhoea through its unique combination of tissue adherence, sealing and fluid absorption. This can potentially prevent lymphatic leak, reduce the drain output and facilitate earlier discharge.

## 2. Materials and Methods

### 2.1. Trial Design

This is a prospective, single-center, phase III randomized controlled trial investigating the efficacy of Hemopatch in preventing lymphatic leak after RARP and BPLND. Participants were randomized to receive RARP and BPLND, with or without the use of Hemopatch. The study was conducted at the Prince of Wales Hospital in Hong Kong from January 2020 to December 2021. The study protocol was approved by the Joint Chinese University of Hong Kong—New Territories East Cluster Clinical Research Ethics Committee (CREC Reference number: 2019.419-T). The study was registered at the US National Institutes of Health (ClinicalTrial.gov; Identifier: NCT04185922). The study was conducted in accordance with the Declaration of Helsinki and the International Conference on Harmonization, Good Clinical Practice Guidelines (ICH-GCP).

### 2.2. Participants

All consecutive patients with prostate cancer, indicated for RARP and BPLND, were screened for eligibility. BPLND was performed if the estimated risk of occult nodal metastases based on the Memorial Sloan Kettering Cancer Center pre-radical prostatectomy nomogram [13] was over 5% [14]. The inclusion and exclusion criteria are as follows.

Inclusion criteria
-Aged 18 years and above-Able to give informed consent-Suitable for minimally invasive surgeryExclusion criteria
-Known allergy or hypersensitivity to any component of Hemopatch-Known hypersensitivity to bovine proteins or brilliant blue-Patients with prior pelvic radiotherapy-Patients with non-correctable coagulopathy-Patients who are on anticoagulants-Contraindication to general anaesthesia-Previous transurethral resection of prostate (TURP) or prostatic surgery-Untreated active infection

Informed consent was obtained from the eligible study subjects before the scheduled RARP and BPLND operations.

### 2.3. Randomization and Blinding

Patients were randomized to receive RARP and BPLND, with or without Hemopatch, with an allocation ratio of 1:1. The randomization sequence was obtained with computer-generated random sequence numbers, with no restriction rules applied. Allocation concealment was ensured by the use of a web-based internet application to reveal randomization codes after patient recruitment. The patients were then assigned to the experimental arm or the standard arm according to the randomization codes. The operating urologist was informed of the allocated treatment arm only after BPLND had been performed. Patients receiving the treatment and investigators assessing the study outcomes were blinded from the allocated treatment arm.

### 2.4. Interventions

All participants received RARP and BPLND in the usual manner [15]. Each patient was first placed supine with a split leg position. Skin incisions were made to allow for the insertion of the robotic camera port, robotic instrument ports and assistant ports. The patient was then placed in a reverse Trendelenburg position, and the da Vinci Xi robotic surgical system was docked. Radical prostatectomy was performed with an anterior approach. After the prostate gland was excised, BPLND was performed with a standard template up to uretero-iliac crossing. Vesicourethral anastomosis was then performed, and a water leak test was routinely performed to ensure a water-tight anastomosis. After the vesicourethral anastomosis was completed, four pieces of Hemopatch were applied to the lymph node dissection area on each side, i.e., eight pieces in total. The distal ends at the obturator fossa and the femoral canal, along with the proximal ends at the the common iliac bifurcation and internal iliac artery, were thoroughly covered by Hemopatch (see Figure 1). The patch was kept dry until contact with the tissue. After tissue contact, pressure was applied over the pad surface for two minutes. The active comparator is standard RARP and BPLND without the use of haemostatic adjuncts, which is the standard of care at our institution. Finally, a pelvic drain was inserted before the conclusion of the operation. 

### 2.5. Post-Operative Management

The diet was usually resumed on post-operative day 1. The pelvic drain would be removed if the output was <100 mL over the past 24 h. The patient would be discharged with a Foley catheter when the diet was well tolerated and drain was removed. The patient would return to the hospital for the removal of the urethral catheter on post-operative days 7 to 10, and this was subjected to the discretion of the operating surgeon. A further follow-up appointment was arranged one month after the operation.

### 2.6. Outcome Measures and Data Collection

The primary outcome is the total volume of drain output, which is a surrogate for lymphorrhoea. The hypothesis is that Hemopatch placement prevents lymphatic leak.

The secondary outcomes include the estimated blood loss, operative time, lymph node yield, duration of drainage, drain output per day, hospital stay, transfusion and 30-day complications.

All baseline characteristics and peri-operative complications were recorded upon the day of discharge. Any 30-day complications following the operation were recorded during the follow-up appointment. All complications were graded according to the Clavien–Dindo classification. Serious adverse events were reported until 30 days after the operation. 

### 2.7. Sample Size

A total of 32 participants were recruited for the study, as per study protocol. Assuming a 33.3% difference in the total volume of drain output (100 mL in the Hemopatch group vs. 150 mL in the control group, with a standard deviation of 50 mL), with a two-sided significance of 0.05 and a power of 80%, 16 patients are required in each group. As the primary outcome is the total volume of drain output, which will be determined upon hospital discharge, no drop-out rate is included in the sample size calculation.

### 2.8. Statistical Methods

All outcome measurements were analyzed with an intention-to-treat principle. An independent samples t-test was used for parametric continuous variables; a Mann–Whitney U test was used for non-parametric continuous variables; a chi-square test was used for categorical variables. A *p*-value less than 0.05 is considered to be statistically significant. All statistical analyses were performed using SPSS version 23.0 (Armonk, NY, USA: IBM Corp.).

## 3. Results

### 3.1. Overview

From 28 February 2020 to 2 July 2021, 36 patients were screened for study eligibility, and 32 participants were recruited into this study. All participants were randomized and allocated into the treatment arm and the control arm, with an allocation ratio of 1:1 (Hemopatch arm: *n* = 16 vs. Control arm: *n* = 16). All participants received their allocated treatments and were followed up in our clinic after the operation. All participants were included in the final analysis. All analyses were performed by the original assigned groups. There were no losses after randomization or losses to follow-up. Figure 2 shows the CONSORT flow diagram.

### 3.2. Patient and Disease Characteristics

The demographic and clinical characteristics of the two arms are summarized in Table 1. The mean age, weight, height and ASA grading were similar between the two groups. The baseline serum PSAs were similar: 9.0 ng/mL in the control group and 10.5 ng/mL in the Hemopatch group. The control group patients had a mean prostate volume of 53.1 cm^3^ compared to 41 cm^3^ in the Hemopatch group. A total of 56.3% and 68.8% of the control group and the Hemopatch group had prostate cancer with an ISUP grade group ≥ 3, respectively, and 62.5% and 56.3% of the respective arms had high-risk disease. A total of 56.3% participants in both arms had pT3 disease, and only one (6.3%) patient from the control arm was staged with pN1 disease.

### 3.3. Study Outcomes

In terms of primary outcome, the median total volumes of the drain output were 180 mL (IQR: 73–558) in the control group and 35 mL (IQR: 1–190) in the Hemopatch group. The total drain output is statistically significantly lower in the Hemopatch group (*p* = 0.022). For secondary outcomes, the Hemopatch group also demonstrated a statistically significantly lower drain output volume per day, with a median drain output per day of 35 mL/day (IQR 1–117) in the Hemopatch group compared to 89 mL/day (IQR 68–139) in the control arm (*p* = 0.038). The median duration of drainage was 2 days in the control arm and 1 day in the Hemopatch arm, and the mean duration of hospital stay was 5 days in the control arm and 3 days in the Hemopatch arm. Both drainage duration and hospitalization duration were slightly shorter in the Hemopatch arm; however, the differences did not achieve statistical significance. There are also no statistically significant differences in terms of operation time, lymph node yield, intra-operative blood loss or post-operative 30-day complications. None of the patients required transfusion post-operatively. Table 2 summarizes the primary and secondary outcome findings.

In the control group, five patients (31.3%) experienced complications within 30 days after operation, all being of Clavien–Dindo grade 1. These complications include right groin numbness, perineal discomfort, drain leakage, urine leak and prepuce oedema. All were conservatively managed. In the Hemopatch group, three patients (18.8%) experienced complications within 30 days, with one patient experiencing two complications. Two patients developed a Clavien–Dindo grade 2 complication of wound infection that required antibiotics use and dressing. The rest of the complications were of Clavien–Dindo grade 1, including fever and paraphimosis. The results are summarized in Table 3.

## 4. Discussion

Prostate cancer is the second most common cancer affecting men globally [16], with an incidence rate that is rising with time [17]. Radical prostatectomy is recognized to be a standard modality of treatment that improves cancer-specific survival [18], and same-session BPLND is performed for those with an estimated risk of occult nodal metastases exceeding 5%, which provides valuable staging and prognostic information that cannot be matched by other available procedures [2]. However, BPLND is not without risks, with lymphocoele formation being the most common complication caused by the disruption of lymphatic drainage after pelvic lymph node dissection. Lymphocoele formation can cause compressive symptoms and infective complications and negatively affect the post-operative recovery of prostatectomy patients. Various techniques to improve lymphostasis, including bipolar diathermy, clip application [19], peritoneal flap interposition [20] and haemostatic pad application with TachoSil [21], have been explored, with no definite data to suggest the best approach.

Hemopatch is considered the second generation of advanced hemostatic pads [6]. The effectiveness of Hemopatch is largely attributed to its physical properties, which consist of a sheet-like collagen backing with a self-binding surface, with the binding agent being NHS-PEG. The design allows for a dual mechanism that allows for the rapid adherence of tissue and fluid from electrophilic cross-linking with the NHS-PEG monomers, while the collagen scaffolding would mediate intrinsic hemostatic action to form fibrin clots. By laying Hemopatch over raw lymphatic tissue, the highly porous bovine collagen sheet would allow for rapid tissue fluid absorption and increase the surface area for the delivery of NHS-PEG for sealant functions. 

This is the first randomized controlled trial on the use of Hemopatch, a novel haemostatic agent composed of a bovine collagen sheet coated with NHS-PEG, in the setting of RARP with BPLND for prostate cancer patients. In this trial, the outcomes demonstrated that the application of Hemopatch to raw lymphatic tissues could reduce lymphorrhoea as shown by the lower total drain output volume and drain output volume per day when compared to the standard procedure without the use of haemostatic adjuncts. The median total drain output is only 35 mL in BPNLD with the use of Hemopatch, compared to 180 mL in the control group, showing a more-than-five-times reduction in output volume. The operative times and 30-day complications were similar between the two groups. These findings suggest that Hemopatch is a safe and effective haemostatic agent that can be laid over the ends of raw truncated lymphatic tissue after BPLND to reduce lymphorrhoea and post-operative drain output, with no additional morbidity or complication.

Hemopatch has been utilized across multiple surgical specialties [22]. In urology, a previous prospective study investigated the use of Hemopatch in laparoscopic partial nephrectomy [23,24], demonstrating its ability to help achieve haemostasis. The prospective series explored the use of Hemopatch by a single surgeon in 19 patients receiving laparoscopic partial nephrectomies (16 with zero-clamping, zero-ichemia), which showed that the application of Hemopatch successfully achieved hemostasis in all cases. However, no studies have investigated the role of Hemopatch in reducing or peventing lymphorrhoea, let alone in the RARP and BPLND setting. One randomized controlled trial of 100 patients investigated the use of TachoSil [21], a hemostatic patch comprised of a collagen sponge coated with fibrinogen and thrombin, in reducing lymphocoele formation after pelvic lymph node dissection in prostate cancer patients. The results demonstrated statistically significantly less post-operative lymphocoele on cross-sectional imaging in the TachoSil group compared to the control group.

Our study is the first to investigate the role of Hemopatch in the setting of RARP and BPLND and provides good evidence for its use given the randomized controlled design and the quasi-double blinded nature (the surgeon is left blinded for the majority of the operation until randomization). It is also the first study to demonstrate the effectiveness of Hemopatch in reducing lymphorrhoea. Another major strength of this study is the standardization of the procedure, where the Hemopatch placement position (distal ends are at the obturator fossa and the femoral canal; the proximal ends overlie the common iliac bifurcation and internal iliac artery), size and margins (Hemopatch would overlap the margins of the raw area by about 1 cm), application methods (the patch is kept dry until contact with tissue) and pressure application duration (two minutes) are clearly defined and followed. This has reduced bias and inter-surgeon variability when it comes to the application of Hemopatch.

Several limitations exist in this study. First, this is a trial conducted at a single center only; this might have affected the generalizability of the findings. Second, the sample size of the study remained small, with only 16 patients recruited into each arm, which weakened the statistical power, especially in the analysis of secondary outcomes. We have observed that both the drainage duration and hospitalization duration were shorter in the Hemopatch arm; however, the differences were not statistically significant. There are likely true differences in these areas in addition to the drain output; however, our study is underpowered to detect it. Third, our study selected drain output as a surrogate for lymphorrhoea and only followed up patients for 30 days post-operatively. This may not accurately reflect the effectiveness of Hemopatch in reducing the incidence of lymphocele formation after RARP and BPLND. The need for intervention for lymphocele was also not evaluated. To address this, future studies with larger sample sizes and longer follow-ups are needed to evaluate the relationship between Hemopatch application and lymphocele incidence. Another research gap that this study did not address is how Hemopatch performs compared to other currently available adjunctive haemostatic agents, and further study in this area is needed.

## 5. Conclusions

In conclusion, our study is the first randomized controlled blinded trial to demonstrate the effectiveness of Hemopatch, an NHS-PEG coated patch, in reducing the drain output volume after RARP and BPLND in prostate cancer patients. Therefore, the application of Hemopatch to raw lymphatic surfaces should be considered during RARP and BPLND.

## Figures and Tables

**Figure 1 cancers-14-04476-f001:**
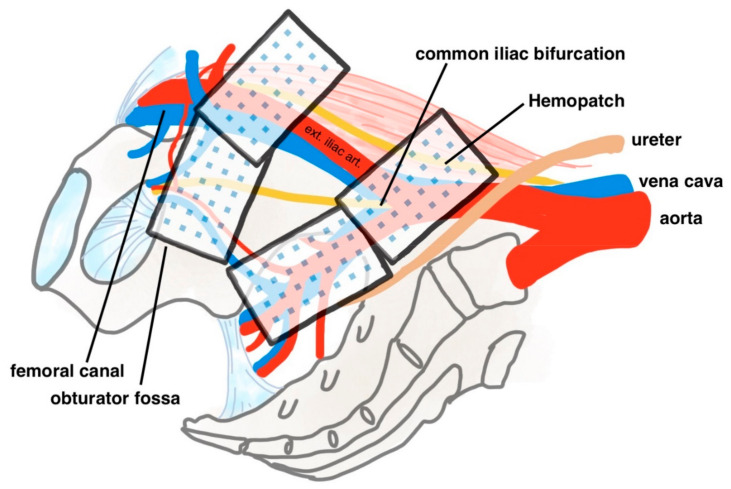
Hemopatch placement position diagram. Ext. iliac art. = external iliac artery.

**Figure 2 cancers-14-04476-f002:**
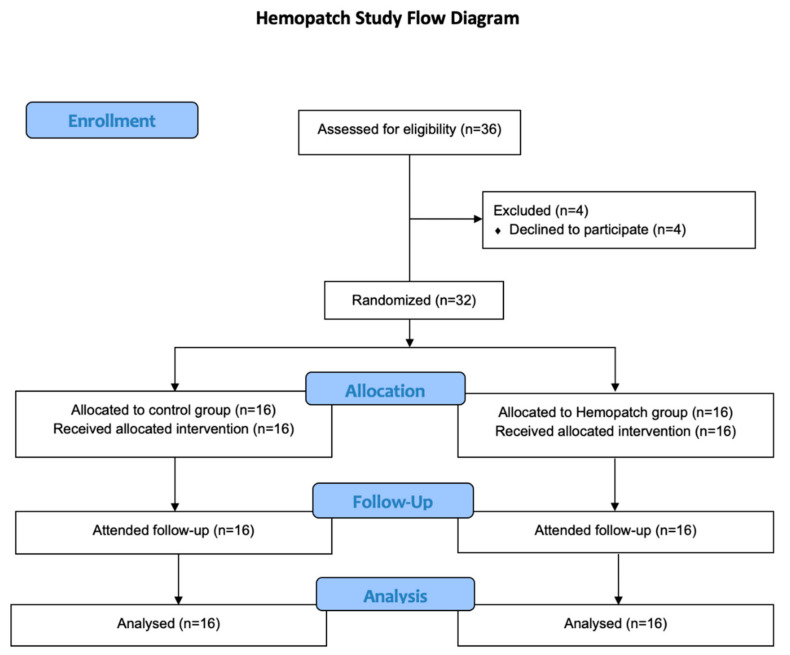
CONSORT flow diagram.

**Table 1 cancers-14-04476-t001:** Baseline demographic and clinical characteristics.

	Control Group (*n* = 16)	Hemopatch Group (*n* = 16)
Age (year)	69 (5)	65 (6)
Weight (kg)	68.2 (10.0)	68.4 (10.2)
Height (cm)	170 (7)	170 (4)
ASA group		
ASA 1	2 (12.5%)	1 (6.3%)
ASA 2	9 (56.3%)	14 (87.5%)
ASA 3	5 (31.3%)	1 (6.3%)
ASA 4	0 (0%)	0 (0%)
Prostate volume (cm^3^)	53.1 (25.8)	41.0 (22.4)
PSA (ng/mL)	9.0 (6.5)	10.5 (6.4)
ISUP ≥ 3	9 (56.3%)	11 (68.8%)
Risk category		
Low-risk disease	0 (0%)	0 (0%)
Intermediate-risk disease	6 (37.5%)	7 (43.8%)
High-risk disease	10 (62.5%)	9 (56.3%)
cT stage		
cT1	9 (56.3%)	12 (75.0%)
cT2	7 (43.8%)	4 (25.0%)
pT stage		
pT2	7 (43.8%)	7 (43.8%)
pT3a	3 (18.8%)	6 (37.5%)
pT3b	6 (37.5%)	3 (18.8%)
pN1	1 (6.3%)	0 (0%)

Continuous variables are presented as the mean (SD). Categorical variables are presented as *n* (%). ASA = American Society of Anaesthesiology; ISUP = International Society of Urological Pathology; PSA = Prostate-Specific Antigen.

**Table 2 cancers-14-04476-t002:** Primary and secondary outcomes.

	Control Group (*n* = 16)	Hemopatch Group (*n* = 16)	*p*-Value
**Primary outcome**			
Total drain output (mL) *	180 (73–558)	35 (1–190)	0.022
**Secondary outcomes**			
Operative time (minute)	189 (45)	175 (52)	0.449
Intra-operative blood loss (mL)	272 (244)	209 (156)	0.395
Number of lymph nodes excised	13.6 (6.6)	12.69 (4.4)	0.663
Duration of drainage (day) *	2 (1–5)	1 (1–2)	0.139
Drain output per post-op day (mL/day) *	89 (68–139)	35 (1–117)	0.038
Hospital stay (day)	5 (3)	3 (1)	0.105
Transfusion	0 (0%)	0 (0%)	-
30-day complications	5 (31.3%)	3 (18.8%)	0.685

* Continuous variables are presented as means (SD) unless otherwise specified. * Continuous variables are presented as medians (IQR) and compared by the Mann–Whitney U test. Categorical variables are presented as *n* (%) and compared by Fisher’s exact test. A two-sided *p*-value of <0.05 is considered statistically significant.

**Table 3 cancers-14-04476-t003:** Summary of 30-day complications.

	Control Group (*n* = 16)	Hemopatch Group (*n* = 16) *
Clavien–Dindo grade 1	5 (31.3%) i. Drain leakage (*n* = 1, 6.3%) ii. Perineal discomfort (*n* = 1, 6.3%) iii. Prepuce edema (*n* = 1, 6.3%) iv. Right groin numbness (*n* = 1, 6.3%) v. Urine leak (*n* = 1, 6.3%)	1 (18.8%) i. Fever (*n* = 1, 6.3%) ii. Paraphimosis (*n* = 1, 6.3%)
Calvien–Dindo grade 2	0 (0%)	2 (12.5%) i. Surgical site infection (*n* = 2, 12.5%)

* One patient from the Hemopatch arm was complicated with paraphimosis and wound infection.

## Data Availability

The data presented in this study are available on request from the corresponding author. The data are not publicly available due to the data containing patients’ clinical information.

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
