# Peer review of "Hemopatch to Prevent Lymphatic Leak after Robotic Prostatectomy and Pelvic Lymph Node Dissection: A Randomized Controlled Trial"

_cancers, 2022, doi:10.3390/cancers14184476_

Round 1
Reviewer 1 Report
Nice paper on the merits of a hemostatic agent to prevent lymph drainage in patients who undergo RALP and lymfnode dissection.
Small study group. I wonder whether a sample size calculation has been performed before the study was initiated and what were the expected outcomes.
The major question remains, what is the goal to decrease lymph drainage? This should be mentioned by the authors. The surgeons place a drain for what reason? In most series a drain is only placed for urine leakage or to know whether there is a bleeding (in difficult cases). So, get out the drain when it is no urine and whatever the production is (100 or 1000 mL)
So just leave out the drain?
Author Response
Thank you for your kind comments.
Regarding your first query on the sample size – yes, we have indeed calculated the sample size before the initiation of the study. Assuming a 33.3% difference in the total volume of drain output (100mL in the Hemopatch group vs. 150mL in the control group with a standard deviation of 50mL) with a two-sided significance of 0.05 and a power of 80%, 16 patients will be required in each group. As the primary outcome is the total volume of drain output which will be concluded upon hospital discharge, no drop-out rate is included in the sample size calculation. Therefore, our plan was to recruit 32 patients in total, which we have achieved in the final study.
In this study, the drain output is a surrogate for lymphorrhoea after bilateral pelvic lymph node dissection. Our hypothesis is that the application of Hemopatch to raw lymphatic tissue can prevent lymphorrhoea through its unique combination of tissue adherence, sealing and fluid absorption. This can potentially prevent lymphatic leak, reduce drain output and facilitate earlier discharge. Therefore the drain placement, apart from the clinical function for urine leakage or bleeding, is to assess the output in the surgical site. We have included the aforementioned goal in the introduction section, and have updated the methodology section to clarify these points. Thank you for your suggestions.
Reviewer 2 Report
This is a study with implications for guiding clinical practice and the results provided by the authors so far are promising. However, there are several issues in this study that I am very concerned about.
1. The number of cases enrolled in this clinical trial in this study is too small and I am not sure that this meets the relevant requirements.
2. The duration of observation was also relatively short, which may have missed the occurrence of some chronic complications.
3. A similar product as a parallel control, such as the older generation NHS-PEG products mentioned by the authors in the article, would have made the results of this study more convincing.
4. Figure 2 needs to be redrawn, some words are obscured and some formatting needs to be adjusted.
5. The table format is not the same as the journal requires, it should be a three-line table.
Author Response
Thank you for your kind comments.
- We have indeed calculated the sample size before the initiation of the study. Assuming a 33.3% difference in the total volume of drain output (100mL in the Hemopatch group vs. 150mL in the control group with a standard deviation of 50mL) with a two-sided significance of 0.05 and a power of 80%, 16 patients will be required in each group. As the primary outcome is the total volume of drain output which will be concluded upon hospital discharge, no drop-out rate is included in the sample size calculation. Therefore, our plan was to recruit 32 patients in total, which we have achieved in the final study.
- We agree that we may miss complications beyond 30 days and we have addressed this limitation in the discussion section. Regarding the duration of observation, the relatively short duration is appropriate for our selected outcomes, as the lymphorrhoea amount is observed in the initial post-operative period with the drain in-situ, and the assessment of any post-operative 30-day complication should allow the detection of relevant complications such as lymphocoele or collection that would have presented within the observation period.
- Thank you for the suggestion. The goal of this current study is to compare the addition of Hemopatch against the current standard surgical technique (without additional adjunctive agents over lymph node dissection site). The comparison against the older generation NHS-PEG products would be interesting, but it was not included in the study protocol and is out of the scope of this study. It is an interesting idea for future studies.
- Thank you for the observation. We have redrawn Figure 2 and updated the figure and manuscript.
- We have also re-formatted the table into a three-line table as per the journal requirement, thank you.
Reviewer 3 Report
The Authors performed a RCT comparing two groups of radical prostatectomy patients. They used Hempatch in one of the two arms while in the second, control arm, they did not. The Authors demonstrated a significant reduction of lymphatic leak in the active arm.
The paper is well written and methods are correct. Blinding sufficiently warrented. Patients cohort is small and results should be verified in multi-institutional study
Author Response
Thank you for your kind comments. Indeed the sample size of the study is small, but based on our sample size calculation it is sufficient to detect our outcome of interest. A larger scale multi-institutional design for verification can be considered downstream. I have included the sample size calculation rationale below for your reference.
Assuming a 33.3% difference in the total volume of drain output (100mL in the Hemopatch group vs. 150mL in the control group with a standard deviation of 50mL) with a two-sided significance of 0.05 and a power of 80%, 16 patients will be required in each group. As the primary outcome is the total volume of drain output which will be concluded upon hospital discharge, no drop-out rate is included in the sample size calculation. Therefore, our plan was to recruit 32 patients in total, which we have achieved in the final study.
Round 2
Reviewer 2 Report
The author's revision is satisfactory.